# Spot-fire distance increases disproportionately for wildfires compared to prescribed fires as grasslands transition to *Juniperus* woodlands

**Victoria M. Donovan**[1,2]*, **Dillon T. Fogarty**[1], **Dirac Twidwell**[1]

**1** Department of Agronomy and Horticulture, University of Nebraska-Lincoln, Lincoln, Nebraska, United States of America, **2** School of Forest, Fisheries, and Geomatics Sciences, West Florida Research and Education Center, University of Florida, Milton, Florida, United States of America

* victoria.donovan@ufl.edu

**Data Availability Statement:** Data is available on Dryad via https://doi.org/10.5061/dryad.vmcvdncxs.

## Abstract

Woody encroachment is one of the greatest threats to grasslands globally, depleting a suite of ecosystem services, including forage production and grassland biodiversity. Recent evidence also suggests that woody encroachment increases wildfire danger, particularly in the Great Plains of North America, where highly volatile *Juniperus* spp. convert grasslands to an alternative woodland state. Spot-fire distances are a critical component of wildfire danger, describing the distance over which embers from one fire can cause a new fire ignition, potentially far away from fire suppression personnel. We assess changes in spot-fire distances as grasslands experience *Juniperus* encroachment to an alternative woodland state and how spot-fire distances differ under typical prescribed fire conditions compared to conditions observed during wildfire. We use BehavePlus to calculate spot-fire distances for these scenarios within the Loess Canyons Experimental Landscape, Nebraska, U.S.A., a 73,000-ha ecoregion where private-lands fire management is used to reduce woody encroachment and prevent further expansion of *Juniperus* fuels. We found prescribed fire used to control woody encroachment had lower maximum spot-fire distances compared to wildfires and, correspondingly, a lower amount of land area at risk to spot-fire occurrence. Under more extreme wildfire scenarios, spot-fire distances were 2 times higher in grasslands, and over 3 times higher in encroached grasslands and *Juniperus* woodlands compared to fires burned under prescribed fire conditions. Maximum spot-fire distance was 450% greater in *Juniperus* woodlands compared to grasslands and exposed an additional 14,000 ha of receptive fuels, on average, to spot-fire occurrence within the Loess Canyons Experimental Landscape. This study demonstrates that woody encroachment drastically increases risks associated with wildfire, and that spot fire distances associated with woody encroachment are much lower in prescribed fires used to control woody encroachment compared to wildfires.

**Funding:** DT received funding Nebraska Game & Parks Commission (W-125-R-1) http://outdoornebraska.gov/ Nebraska Environmental Trust (20-139-2) https://environmentaltrust.nebraska.gov/ University of Nebraska's Institute of Agriculture and Natural Resources https://ianr.unl.edu/ The funders had no role in study design, data collection and analysis, decision to publish, or preparation of the manuscript.

**Competing interests:** The authors have declared that no competing interests exist.

## Introduction

Woody encroachment is one of the greatest threats to grasslands globally [1–6], increasing the level of degradation to one of the world's most at-risk ecosystems [7,8]. As a system transitions from a grassland state to a woody state, rapid and persistent changes in system processes and structure occur [9–11]. For instance, woody encroachment can alter grassland properties by shifting a system from below to above ground biomass storage [12], decreasing grassland species richness and diversity [3,13], altering soil carbon and nitrogen [14], decreasing herbaceous biomass production [15,16], and increasing demands on the water table [17]. Changes in fuel structure during the transition from a grassland to woodland state also shift fire regime characteristics from frequent, low flame length surface fires in grasslands to infrequent, high flame length crown fires in woodlands [11,18], with recent evidence suggesting that this transition can increase large wildfire risk [19].

Given recent trends of increasing wildfire activity and woody encroachment in North America [4,20,21], there is a need to quantify relationships between fire risk and woody encroachment so land managers can make informed decisions tied to fire management and planning. In regions like the North American Great Plains, woody encroachment can largely be tied to the loss of frequent fire in grassland systems due to fire suppression and the loss of human fire stewardship practices [1,22]. The application of prescribed fire in grassland and woodland states can prevent and reverse woody encroachment [23–25]. For instance, using high intensity fire to burn juniper woodlands has been shown to overcome hysteretic thresholds and return a system to grassland dominance [23]. However, prescribed fire has not been widely implemented across North America [26,27]. Fire is often viewed as unnatural and risky outside of the scientific community [28–30]. Perceived risks associated with prescribed fires often outweigh that of woody encroachment, meaning that many land managers are more likely to allow woody encroachment than apply prescribed fire [27,31]. However, the increase in woody vegetation in the Great Plains has been associated with recent surging wildfire activity, with risk of large wildfire increasing once woody cover exceeds 20% of the landscape [19,21]. Since it is clear that fire cannot be removed from flammable landscapes [32,33], there is a need to quantify the relative risk of wildfire versus the risk of prescribed fire to control woody encroachment.

One of the greatest concerns associated with managing fire on a landscape is spot fire [34–36]. Spot fires are fires that start outside of the original fire perimeter from lofted sparks or firebrands and can act as a significant mechanism of fire spread [35]. Fire brands can overcome typical methodologies of fire risk reduction like fuel breaks and remain a primary reason why many private and public land managers are hesitant to conduct prescribed burns [30,34]. Escaped fires can be disastrous, resulting in the destruction of property and loss of life [37,38]. Long range spotting that carries embers far outside of the immediate fire area can be more difficult to monitor, control, and suppress. Spotting has played a major role in destructive wildfires [36]. Greater maximum potential spotting distance can increase fire spread rate and fire escape, enhancing the probability for property damage and loss of life.

In this study, we use spot fire distance to examine the relative risk of using prescribed fire to control woody plants at various stages of the encroachment process versus waiting for wildfire to occur. Spot fires are dependent on sequential mechanisms of generation, transport, and ignition in receptive fuels (i.e. fuels that can ignite from a firebrand and support the spread of wildland fire) [35]. During the process of woody encroachment, we predict that the shift towards a woodland will increase the risk of spot fire by enhancing the probability of fire brand generation and lofting. Because of the restricted wind speeds under which prescribed

fires occur, we predict that prescribed fires will have lower maximum spot fire distances, regardless of encroachment stage, and thus, have lower risk potential compared to wildfire.

We use the Loess Canyons Experimental Landscape in Nebraska as a model landscape for our assessment. This region has experienced substantial encroachment from eastern redcedar (*Juniperus virginiana*), a prevalent native invader in the Great Plains that overruns grasslands in the absence of controlling processes like fire [11,22,39]. Prescribed fires are being used to prevent woody encroachment in remaining grasslands in the Loess Canyons, but because of the extent of encroachment already prevalent on the landscape, landowners are also using prescribed fire to remove established juniper trees in invaded areas to revert them back to grasslands [23,40]. Quantifying the relative risks of employing prescribed fire as a controlling process for woody encroachment versus waiting for wildfire to occur will help land mangers make informed decisions regarding the application of prescribed fire to manage woody encroachment.

## Materials and methods

### Study area

The Loess Canyons Experimental Landscape is a 73,000-ha area of mixed grass prairie located in south-central Nebraska (Fig 1). The landscape represents a private-lands approach to science co-production [41] where partnerships among landowners, natural resource agencies, and scientists support landowner-led efforts to confront a regional trend of woody encroachment [40]. Vegetation communities can be described based on a gradient of woody encroachment from un-encroached mixed grass prairie to a juniper woodland state that has resulted from unchecked woody encroachment into grassland areas. Prior to European settlement, the region experienced frequent fire (6–10 year fire return interval) [42] and was maintained as a prairie region with minimal tree cover. However, fire exclusion has resulted in woody encroachment and the widespread establishment of juniper woodlands. Canyons are the primary landform, with elevation ranging from 781–989 m above sea level. Mean annual precipitation is 550 mm, with the majority of rainfall occurring in the growing season. Mean annual temperature is 9.8˚C (climate data is from [43]).

Between 2002 and 2019, over 90 prescribed burns have been conducted in the Loess Canyons Experimental Landscape to assist in regional-scale grassland restoration [23]. Prescribed burns were conducted by local prescribed burn associations and ranged in size from 9–1,041 ha. Burns are typically conducted between early February and late April and target fire intensities that exceed juniper mortality thresholds [46]. Fuel manipulation is used to increase prescribed fire intensity. Prescribed burns in the Loess Canyons Experimental Landscape therefore result in high rates of woodland collapse and stimulate herbaceous plant recovery [23]. The Loess Canyons is currently in the reclamation phase of restoration, and thus, a prescribed fire return interval has yet to be established in this region.

### Maximum spot fire distance models

To predict spot fire behaviour, we calculated maximum spot fire distance using BehavePlus software v. 5.0.5. BehavePlus is a mathematical fire simulation program that predicts fire behaviour relative to environmental and fuel-based characteristics using a point system, where conditions are assumed uniform. It is widely used by fire managers in the U.S. to predict fire behaviour and assess wildland fuel hazards [47]. The SPOT module in BehavePlus calculates maximum spot fire distance based on models developed by Albini [48–50] for both surface fire and torching trees [51].

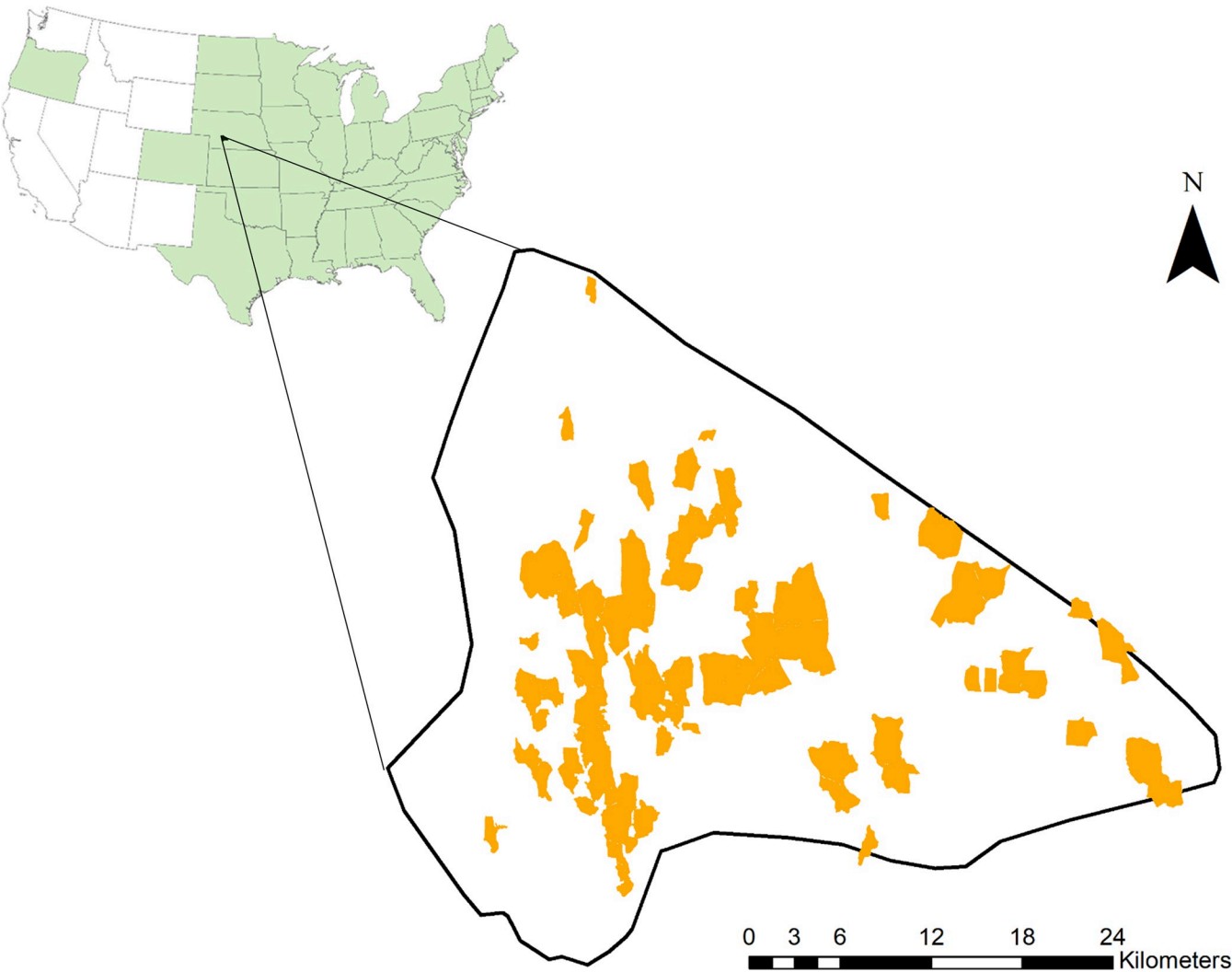

**Fig 1. The distribution of prescribed fires (yellow) from 2000–2019 in the Loess Canyons Biologically Unique Landscape in Nebraska.** The map inset shows the location of the Loess Canyons in the continuous USA, taken from the U.S. Department of Commerce [44]. States where *Juniperus virginiana* occurs are shaded green based on the USDA-NRCS PLANTS Database (https://plants.usda.gov/). Maps were generated using ArcGIS [45].

We used BehavePlus to compare the relative spot fire distance that could occur as a result of fire burning under three different levels of juniper encroachment: grassland, juniper encroached grassland, and juniper woodland. To identify the appropriate fuel model input for BehavePlus spot fire calculations, we mapped the distribution of receptive fuel beds in and around the Loess Canyons using maps of Anderson Fire Behaviour Fuel Models [52]. This data contains 13 standard fire behaviour models that represent standard surface fuel arrangements [53] that can serve as input for fire behaviour and fire spread models in BehavePlus (Table 1; S1 Fig). Non-receptive fuels in this land cover classification included agriculture, water, snow/ice, urban areas, and barren areas.

We represented grasslands with the fuel model FBFM1 (Tables 2 and 3), which represents fuel characteristics for short grass, the most common fuel type in the Loess Canyons based on classification by LANDFIRE's Anderson Fire Behaviour Fuel Models landcover maps (Table 1). Because both surface and canopy fuels can contribute to potential spot fires in

**Table 1. Summary of fuel models and their percent cover within the Loess Canyons.** Fuel models and descriptions are from Anderson, [53]. The percent cover of each fuel model type within the Loess Canyons and a surrounding 5 mile buffer were calculated using data from LANDFIRE [52].

| Fuel Model | Typical Fuel Complex [a] | Description [b] | % Cover within the Loess Canyons [c] |
|---|---|---|---|
| FBFM1 | Grass and grass-dominated: Short grass (1 foot) | Fire spread is rapid and determined by fine continuous herbaceous fuels that are cured and curing, very little shrub or timber present, primarily grasslands and savanna | 15% |
| FBFM2 | Grass and grass-dominated: Timber (grass understory) | Fire spread is through fine, herbaceous fuels though litter and dead down stem wood may contribute to spread, open shrub or timber over story cover 1/3 to 2/3 of area | 54% |
| FBFM3 | Grass and grass-dominated: Tall grass (2.5 feet) | Most intense fire of grass group, high rates of spread, stands average 3 ft tall, 1/3 or more of stand is dead or cured | 6% |
| FBFM5 | Chaparral and shrub fields: Brush (2 ft) | Low intensity fires carried by litter, young shrubs that almost totally cover an area with little dead material | 7% |
| FBFM6 | Chaparral and shrub fields: Dormant brush, hardwood slash | Fire spread through shrub layer and requires moderate winds to maintain flame at shrub height | <0.1% |
| FBFM8 | Timber Litter: Closed timber litter | Slow fire spread through ground fuels with low flame lengths, closed canopy stands with short needle conifers or hardwoods, only under severe weather conditions to fuels pose a fire hazard | 7% |
| FBFM9 | Timber Litter: Hardwood litter | Longer flames and quicker surface fires than FBFM8, leaves in the fall can cause spot fire, dead-down woody material can lead to possible torching, spotting and crowing | <0.1% |
| Non-Burnable | | Urban, agriculture, water, barren | 11% |

[a] Typical fuel complex reported in Anderson, [53].

[b] Description summarised from Anderson, [53].

[c] Percentages calculated from LANDFIRE [52].

encroached grasslands, we modelled maximum spotting distance in juniper encroached grasslands in two ways. First, we modelled surface fire in juniper encroached grasslands using fuel model FBFM2 (Tables 2 and 3), which represents grassy timber understory (Table 1). Second, we modelled a single torching tree (Table 3). Tree species was set as Grand fir since juniper species models were not available at the time this study was conducted. In a study conducted by Blunck et al. [54], grand fir and western juniper species were shown to have similar ember flux (number of embers deposited in a region) relative to Doug-fir and Ponderosa pine. We compared spot fire distances modelled using the surface fire model and torching tree model for encroached grasslands. Because outputs were similar, we used surface fire outputs to represent spot fire distances in encroached grasslands because they were slightly higher at higher wind speeds. We modelled a juniper woodland as 30 torching trees (Table 3), the maximum number of torching trees that could be modelled in the BehavePlus in order to represent a dense juniper woodland state.

Surface spot fire models require a flame length and slope parameter. Flame length was calculated in BehavePlus (Table 2). We set fuel moisture at 7–9% to match the minimum recommend fuel moisture to burn under during a prescribed burn [34] (Table 2). Slope was sampled across the Loess Canyons using 120 random sampling points. We assessed the influence of slope on flame length in BehavePlus, setting the maximum slope value assessed to match the maximum percent slope value recorded in the Loess Canyons (45%). While slope influenced flame length, it had little to no influence on spot fire distance. Thus, we held slope constant at zero in our simulations (Tables 2 and 3).

Both tree height and tree diameter influence spot fire distance. For both of our torching tree scenarios, diameter at breast height was set to the mid-point in the range of DBH reported by Lawson [57] for mature juniper trees (Table 3). Tree height was set at maximum juniper height recorded during sampling in the Loess Canyons [40] (Table 3).

**Table 2. Model parameters for calculating flame length values to be input into surface fire spotting distance models.**

| Model parameter | Value | Justification |
|---|---|---|
| *Grassland State* | | |
| Fuel Model | 1 | Most common grassland fuel model |
| 1-h Fuel Moisture | 7% | Minimum recommend fuel moisture to burn under during a prescribed burn [34] |
| Midflame Wind Speed | 0–52 km/h | Calculated by multiplying 6 m wind speed with a correction factor of 0.4 [55] |
| Slope Steepness | 0% | Tests indicated there was little influence of slopes measured in the Loess Canyons on flame length except under low wind speeds, which resulted in minimal changes to spot fire distance. |
| *Encroached Grassland* | | |
| Fuel Model | 2 | Represents juniper invaded grassland |
| 1-h Moisture | 7% | Minimum recommend fuel moisture to burn under during a prescribed burn [34] |
| 10-h Moisture | 8% | Weir [34] suggests a range of 6–15% fuel moisture for 10-h fuels during prescribed fires. National Wildfire Coordinating Group recommends adding 1–2% fuel moisture content to 1-h fuels to calculate 10-h fuel moisture values [55]. We added 1%. |
| 100-h Moisture | 9% | NRCS [56] suggests a range of 7–20% fuel moisture for 100-h fuels during prescribed fires. National Wildfire Coordinating Group recommends adding 2–4% fuel moisture content to 1-h fuels to calculate 100-h fuel moisture values [55]. We added 2%. |
| Live Herbaceous Moisture | 30% | Minimum possible in BehavePlus |
| Mid-flame Wind Speed | 0–52 km/h | Calculated by multiplying 6 m wind speed with a correction factor of 0.4 [55] |
| Slope Steepness | 0% | Tests indicated there was little influence of slopes measured in the Loess Canyons on flame length except under low wind speeds, which resulted in minimal changes to spot fire distance. |

To determine maximum wind speed for our assessment, we used NOAA (www.climate.gov) wind speed records from North Platte Regional Airport Weather Station (Latitude = 41.1213, Longitude = -100.669) between 2010 and 2020, which indicated that wind gusts could reach up to 137 km/h in the Loess Canyons region. However, 129 km/h was the maximum wind speed that could be modelled by BehavePlus. Thus, we set the maximum wind speed at 129 km/h.

We ran models for maximum spot fire distance with wind speeds ranging from 0–129 km/h over a 1.6 km/h (1 mph) interval. In Nebraska, burn plans need to be approved by local fire chiefs, and thus wind speed limits on prescribed burns vary by plan and fire chief [58]. Best management practice guidelines set maximum wind speeds between 24 and 40 km/h [34,58,59]. We represented prescribed fire conditions as the center point of this range (32 km/h) and below. All wind speeds were considered potential wildfire conditions. Changes in maximum spot fire distance relative to juniper encroachment scenario and wind speed were plotted with R statistical software [60,61].

In addition, receptive fuels were mapped using the LANDFIRE 2014 data set. Non-receptive fuels in the Loess Canyons included water, barren lands, agriculture, and urban areas (Table 1). Receptive fuel included short and tall grass, timber with grassy understory, brush, and hardwood and closed timber litter (Table 1). We used buffers representing maximum spot fire distance under different encroachment and wind speed scenarios around prescribed fire

**Table 3. Fuel model parameters and justification for spot fire distance models.**

| Model parameter | Value | Justification |
|---|---|---|
| *Grassland State* | | |
| Downwind Canopy Height | 0 m | Minimum possible in BehavePlus |
| 20-ft Wind Speed | 0–129 km/h | NOAA wind speed records from North Platte Regional Airport Weather Station indicated wind gusts could reach up to 137 km/h in the Loess Canyons region. 129 km/h was the maximum wind speed that could be modelled by BehavePlus. |
| Ridge-to-Valley Elevation Difference | 0 m | There was no influence of elevation on spot fire distance. |
| Flame Length | 0–2.1 m | Recorded flame length varied slightly relative to wind speed. |
| *Encroached Grassland* | | |
| Downwind Canopy Height | 0 m | Minimum possible in BehavePlus |
| 20-ft Wind Speed | 0–129 km/h | NOAA wind speed records from North Platte Regional Airport Weather Station indicated wind gusts could reach up to 137 km/h in the Loess Canyons region. 129 km/h was the maximum wind speed that could be modelled by BehavePlus. |
| Ridge-to-Valley Elevation Difference | 0 m | There was no influence of elevation on spot distance. |
| Flame Length | 0–9.9 m | Flame length varied based on wind speed. |
| *Woodland State* | | |
| Downwind Canopy Height | 0 m | Minimum possible in BehavePlus |
| Torching Tree Height | 12 m | Maximum tree height recorded during field sampling in the Loess Canyons [40]. |
| Spot Tree Species | Grand Fir | There are no juniper species models in BehavePlus. In a study conducted by [54], grand fir and western juniper species were shown to have similar ember flux (number of embers deposited in a region), relative to Doug-fir and Ponderosa pine. |
| DBH | 45.7 cm | The mid-point in the range of juniper DBH [57]. |
| 20-ft wind speed | 0–129 km/h | NOAA wind speed records from North Platte Regional Airport Weather Station indicated wind gusts could reach up to 137 km/h in the Loess Canyons region. 129 km/h was the maximum wind speed that could be modelled by BehavePlus. |
| Ridge-to-Valley Elevation Difference | 0 | Elevations changes observed in the Loess Canyons had no influence on spot fire distance. |
| Number of Torching Trees | 30 | Maximum number of torching trees allowed in BehavePlus |

burn units in the Loess Canyons to assess how receptive fuel exposure changed with woody encroachment level and wind speed in the Loess Canyons.

## Results

Woody encroachment substantially increased maximum potential spot fire distance in grasslands in the Loess Canyons (Fig 2). Maximum spot fire distance did not exceed 1.4 km even under the most extreme wind conditions in grasslands (129 km/h wind speeds; Figs 2 and 3), exposing an average of 1,559 ha ± 471 SD of receptive fuels to spot fire (Figs 3 and 4). In encroached grasslands, this distance was more than doubled (4.3 km; Fig 2), exposing an average 7,912 ha ± 1,654 SD of receptive fuels to potential spot fire (Figs 3 and 4). Following a transition to a woodland state, the maximum spotting distance more than quadrupled, reaching

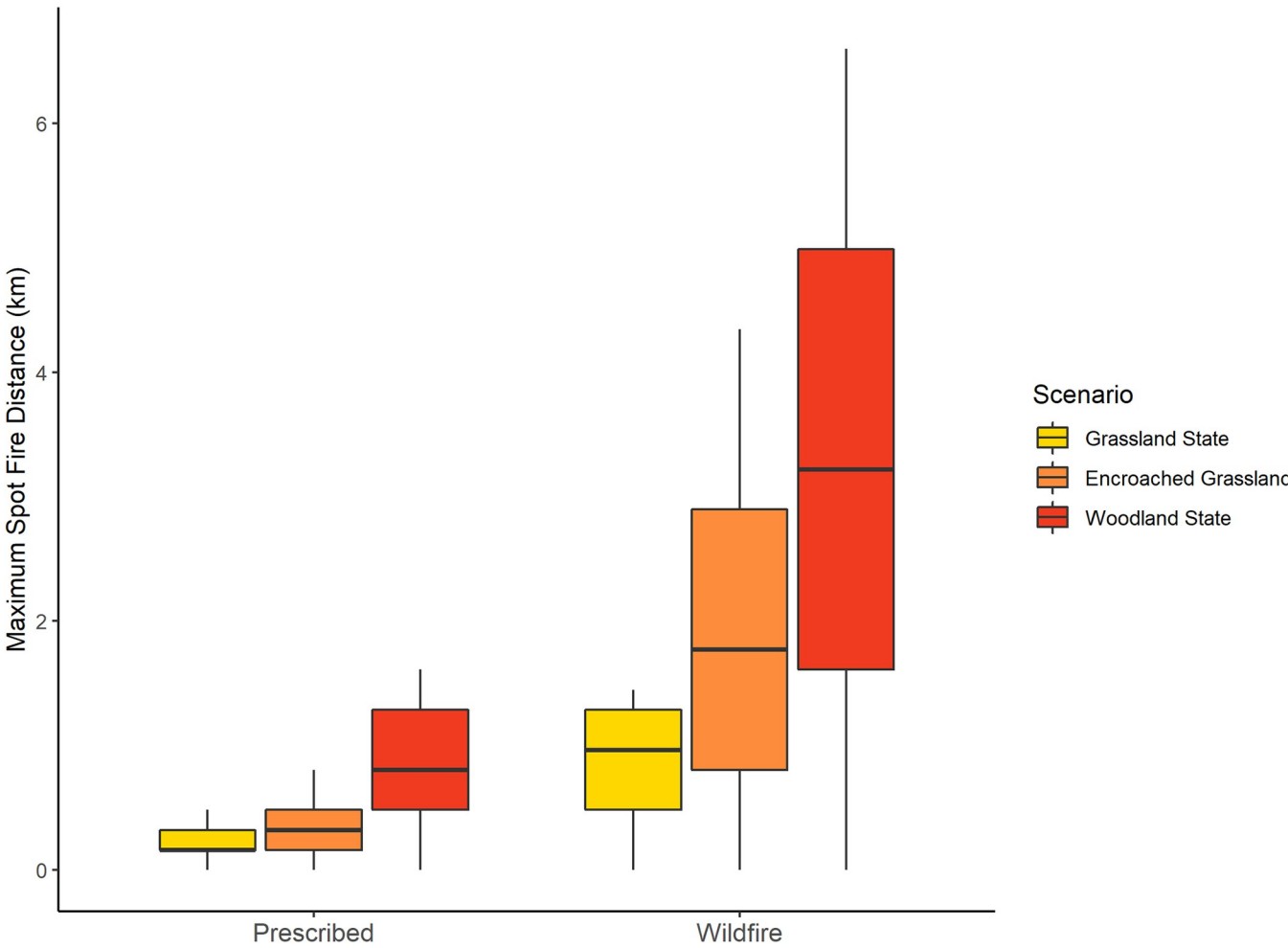

**Fig 2. The difference in maximum spot fire distance under prescribed fire (0–32 km/h) versus wildfire (0–129 km/h) conditions relative to 3 different levels of juniper encroachment: The grassland state, a woody encroached grassland, and the woodland state.**

nearly 7 km (Fig 2), and exposing an average of 15,825 ha ± 2681 SD of receptive fuels to spot fire (Figs 3 and 4).

Fires that burned under prescribed fire wind conditions had lower spot fire risk at each stage of woody encroachment compared to those that burned under the full range of wind conditions. Maximum potential spot fire distances decreased by a factor of ~2 in grasslands under typical prescribed fire conditions (32 km/h wind speeds) compared to high wind speeds that only occur during wildfires (129 km/h; 0.8 km versus 1.4 km; Fig 2). Fires that burn under typical prescribed fire wind speeds also had much lower receptive fuel exposure than fires that burn under wind speeds only seen during wildfire. Receptive fuel exposure was 1,559 ha ± 471 SD under high wind speed conditions (129 km/h) compared to 690 ha ± 388 SD under typical prescribed fire wind conditions in grasslands (Figs 3 and 4). Differences between typical prescribed fire spotting risk versus wildfire spotting risk were further distinguished as woody encroachment progressed. Maximum potential spot fire distance decreased by a factor of ~3 under prescribed fire conditions compared to the high wind speeds only observed during wildfire in encroached grasslands (1.3 km versus 4.3) and *Juniperus* woodlands (6.6 km versus 2.6 km; Fig 2). Moreover, fires that burned under a typical prescribed fire wind speed decreased

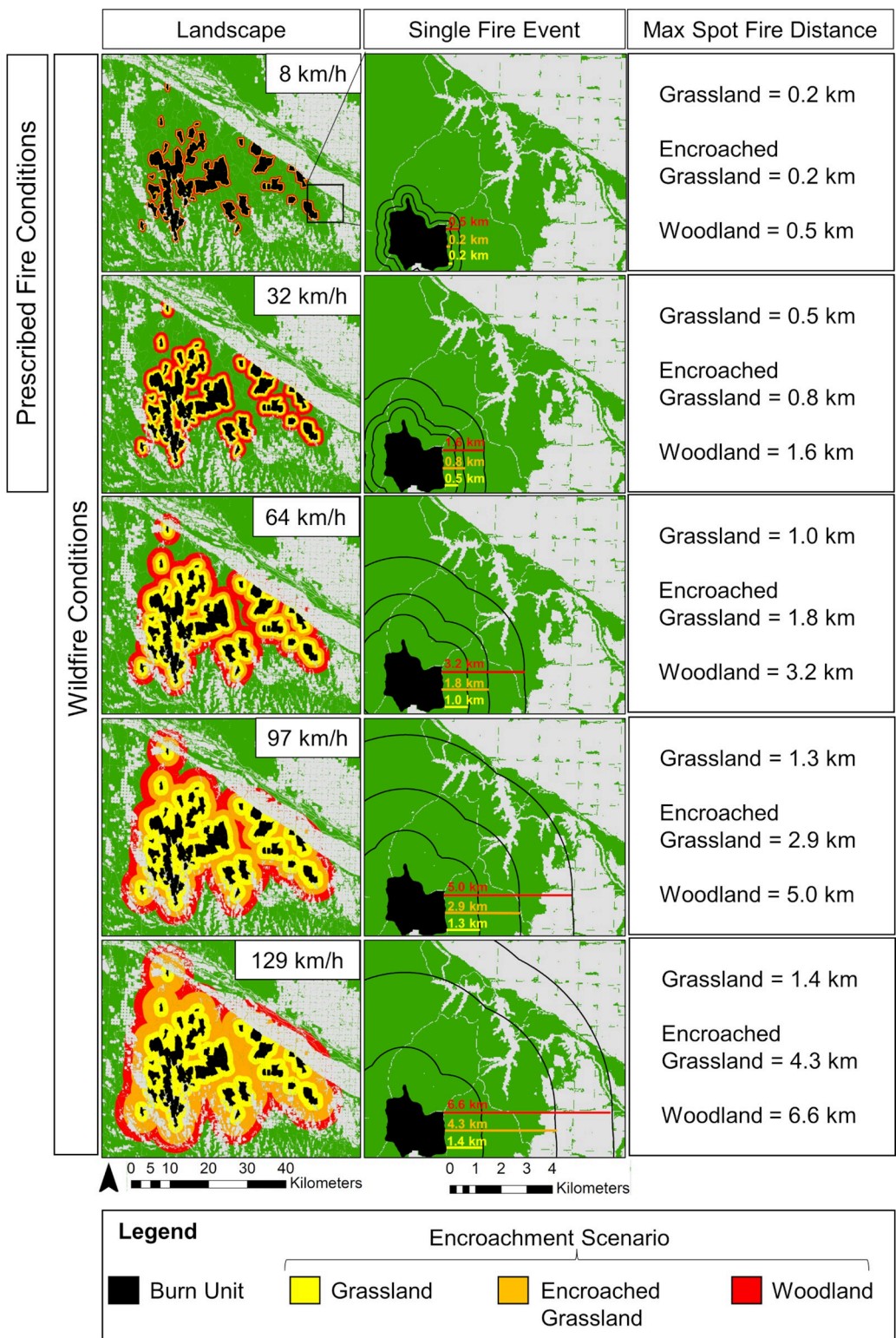

**Fig 3. A comparison of maximum potential spot fire distance under prescribed fire (8 and 32 km/h) versus wildfire (8, 32, 64, 97, and 129 km/h) wind speeds relative to the grassland state (yellow), juniper encroached grasslands (orange), and the juniper woodland state (red) in the Loess Canyons Experimental Landscape.** Column 1 shows a landscape representation of changes in the potential exposure of receptive fuels relative to maximum spot fire distance from burn units in the Loess Canyons if they were all grassland (yellow), all juniper encroached grassland (orange), and

all woodland (red). Column 2 focuses on changes in maximum spot fire distance relative to a single burn unit, where black lines represent the circumference of potential spot fire exposure and coloured lines represent the maximum spot fire distance if the burn unit was all grassland (yellow), woody encroached grassland (orange), or woodland (red). Column 3 lists maximum spot fire distances for each encroachment scenario. Green areas in maps represent receptive fuels and grey areas represent non-receptive fuels (urban areas, agriculture, water, barren areas) [52]. Maps were generated using ArcGIS [45].

average receptive fuel exposure by a factor of over 7 compared to fires that burned under wildfire conditions in encroached grasslands (from 7,912 ha ± 1,654 SD to 1,025 ha ± 489 SD; Figs 3 and 4) and in juniper woodlands (from 2,102 ha ± 749 SD to 15,825 ha ± 2,681 SD).

## Discussion

Prescribed fires have less spot fire risk than wildfires, regardless of the stage of woody encroachment. Spot fire distances under typical prescribed fire conditions were 2- to 3-fold less than those associated with high wind speeds that only occur during wildfires in the same stage of woody encroachment. A key implication from our results is that using prescribed fire

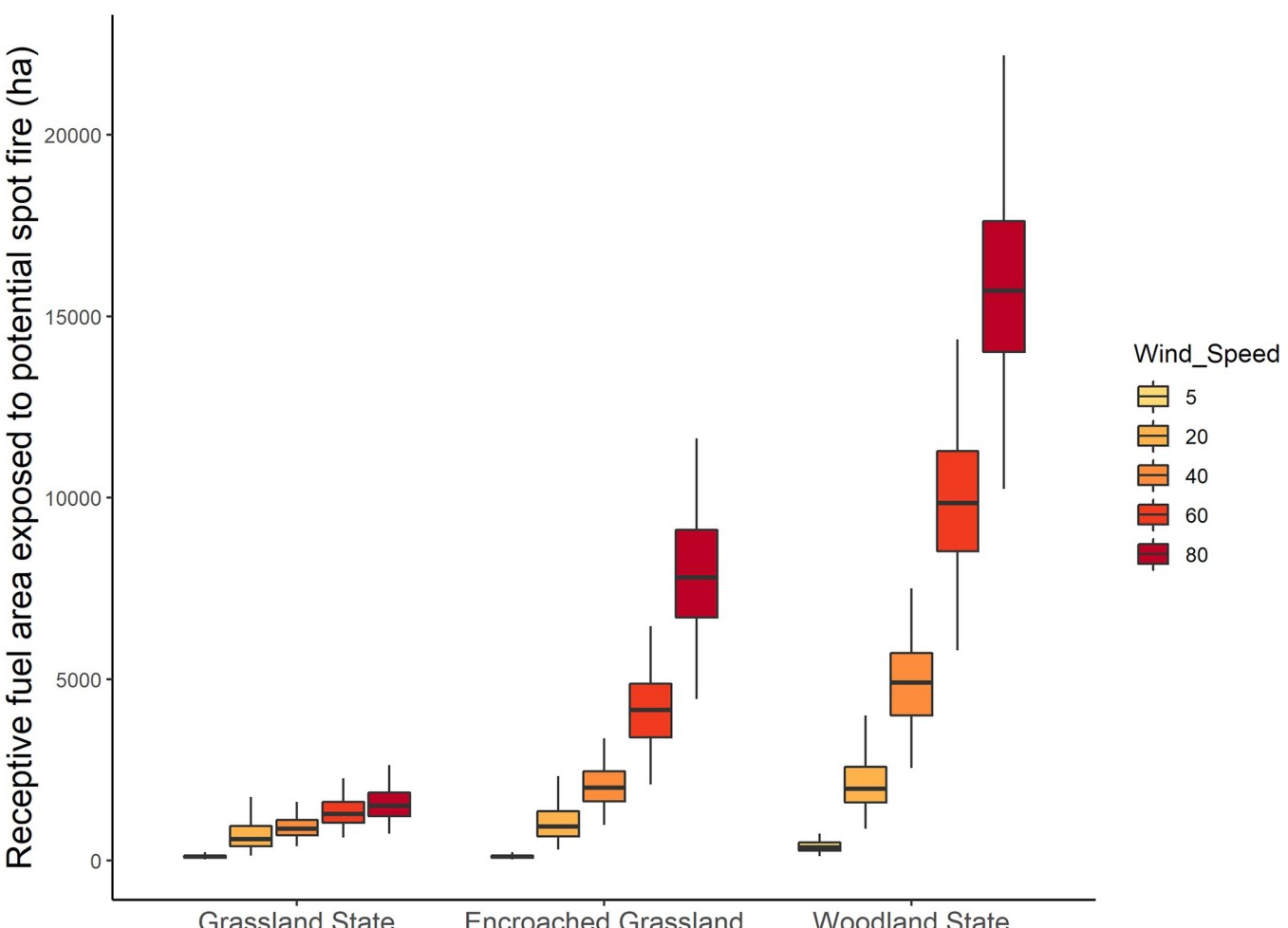

**Fig 4. The change in receptive fuel area exposed to potential spot fire surrounding individual burn units in the Loess Canyons Experimental Landscape as juniper encroachment level and wind speed increase.**

as a controlling process for woody encroachment, either by sustaining existing grasslands or restoring grasslands that have been encroached, has a high potential to reduce fire risk compared to waiting for wildfire to occur. Fire was a critical feedback in grasslands that has not been replaced at large scales and this is compromising our ability to prevent long-term trends of increasing wildfire risk [62,63]. Repeated prescribed fires in grasslands are a well-known and effective method for preventing woody encroachment [4,64,65], while using high intensity prescribed fire in juniper woodlands can reverse woody encroachment [23,46]. For instance, Bielski et al. [23] demonstrate the ability of high intensity fire generated by fuel additions to restore herbaceous biomass following a state-transition to a juniper woodland. However, re-encroachment following high intensity burns in juniper woodlands can begin 1–2 years post-fire and proceeds along a rapid, nonlinear trajectory [15,40]. Thus, frequent low intensity prescribed fires are needed in either maintained or restored grasslands to prevent woody encroachment.

Woody encroachment increases spot fire risk. Maximum spot fire distance greatly increased as sites shifted from a grassland to a woodland state in the Loess Canyons. Woody vegetation is associated with more extreme fire behavior including long flame lengths that can exceed 14 m, making fires that ignite the crown in woody fuels more difficult to control and suppress [11]. It has been speculated that the high propensity for wildfire in woody vegetation in the Great Plains is associated with lower probabilities of fire suppression [19]. Efforts to reduce future wildfire risk should focus on maintaining grassland sites using prescribed fire, which we show have the lowest spot fire risk. Prescribed fires applied after woody encroachment has begun will require higher fire intensities to drive mortality of mature juniper [25,46], while lower intensity fires can be used to control juvenile woody plants. Management can often underinvest at the early stages of encroachment and overinvest in the later stages where control is less likely, more costly, and as we demonstrate, hosts greater risk [39,66,67]. Our results further emphasize that early application of prescribed fires is the best strategy for reducing fire risk in flammable ecosystems like the Great Plains.

Prescribed fires will help reduce the costs of fire management and suppression. Identifying and suppressing spot fire requires sufficient allocation of ground crews, air tankers, Unmanned Ariel Devices (UAVs), or other resources across the potential spot fire range to conduct fire suppression tasks and asset protection [68–70]. We found that woody encroachment substantially increased potential spot fire distance and subsequently the amount of area that needed to be monitored during the application of prescribed fire, indicating there is higher risk associated with applying prescribed burns after encroachment has already begun. However, fire management costs will be further exacerbated by waiting for wildfires to occur, which increased the area of potential spot fire occurrence in our assessment. Because it is clear that fire cannot be removed from flammable landscapes [32,33], applying prescribed fires, regardless of encroachment phase, will decrease demands on suppression personnel and equipment and risks to human life and property. Land managers tend to be more likely to allow woody encroachment than apply prescribed fire [27,31]. However, with both wildfires and woody encroachment increasing across much of North America [20,21,71], continuing to avoid prescribed fire application will strain already limited fire management resources that struggle to keep up with increasing large wildfire occurrence [21,33].

Changing social perspectives and double-think management policies tied to woody vegetation and fire will be imperative to promoting pro-active grassland management [39]. Fire and afforestation policies based on the assumption that fire can be eliminated from landscapes need to be re-evaluated [21,32,33]. Often management policies and social perspectives oppose well established ecological knowledge tied to woody encroachment and fire [26,39]. It is clear that fire is an inevitable terrestrial ecosystem process that maintains and benefits grassland

systems [22,72–75]. Co-existence with fire was achieved through fire stewardship by the Indigenous peoples of North America for thousands of years before Euro-American colonisation [22,75,76]. Elimination of human stewarded fire increases fuel accumulation and woody vegetation, promoting large wildfires outside of human control that can lead to losses of human life and property [19,21]. As we demonstrate, there is higher risk associated with wildfires that burn in woody encroached grasslands. Yet afforestation of grasslands is still largely promoted and subsidised [77–79]. Moreover, prescribed fire application is highly regulated, where fire ban policies, strict liability laws, and lack of private citizen knowledge of prescribed fire limits its use across grasslands [11,31,78,80]. Continued outreach and education regarding the benefits of prescribed fire for reducing wildfire risk associated with woody encroachment is needed.

There are clear social-ecological benefits to applying wide-spread prescribed fire in grassy ecosystems [11,63,81]. Typical brush control methods are associated with significant economic investments that function at too small of a scale to combat the wide-spread woody encroachment occurring in the Great Plains [78]. Intensive control methods for juniper, like herbicides, can cost over $90 per hectare to apply compared to prescribed fire treatments which cost less than $5 per hectare [82]. Prescribed fires can prevent and reverse woody encroachment to mitigate wildfire risk [11,23,46]. In addition, prescribed fires can boost forage production [83], enhance grassland biodiversity and wildlife habitat [84–86], and control invasive species [87,88]. Our results provide an additional level of scientific support for the application of prescribed fire in grassland systems that can be used to support policy and management that promote pro-active prescribed fire application.

## Supporting information

**S1 Fig. The distribution of Anderson [53] fuel models in the Loess Canyons (dark grey outline) from LANDFIRE [52].**
(TIF)

## Acknowledgments

We thank landowners in the Loess Canyons who provided their prescribed burn information.

## Author Contributions

**Conceptualization:** Victoria M. Donovan, Dirac Twidwell.

**Data curation:** Dillon T. Fogarty.

**Formal analysis:** Victoria M. Donovan.

**Funding acquisition:** Dirac Twidwell.

**Methodology:** Victoria M. Donovan, Dillon T. Fogarty, Dirac Twidwell.

**Validation:** Victoria M. Donovan.

**Visualization:** Victoria M. Donovan, Dirac Twidwell.

**Writing – original draft:** Victoria M. Donovan.

**Writing – review & editing:** Victoria M. Donovan, Dillon T. Fogarty, Dirac Twidwell.

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
