## [Decision Letter · Decision Letter 0]

23 Nov 2022

PONE-D-22-20383

Prescribed fire reduces wildfire risk from woody encroachment in grasslands

PLOS ONE

Dear Dr. Donovan,

Thank you for submitting your manuscript to PLOS ONE. After careful consideration, we feel that it has merit but does not fully meet PLOS ONE’s publication criteria as it currently stands. Therefore, we invite you to submit a revised version of the manuscript that addresses the points raised during the review process.

I'm awfully sorry to get back to you after such a long time. Meanwhile, I received comments from one reviewer and additionally assessed the manuscript myself. The reviewer noted that the manuscript should be improved as it lacked clarity in some parts of the text and Figures could be reduced and improved.

Please also find my additional thoughts:

L55: am not sure as to whether the term "extreme fire" is sufficiently clear in this context. Does "extreme" refer to the fire severity?

L103-104: here you mention that vegetation communities are dominated by two "alternative states: mixed grass prairie communities and monoculture woodlands consisting of mature juniper trees".

However, as of L131-132 it seems that there are at least three different levels of juniper encroachment, including vegetation types ranging from grasslands, to juniper encroached grasslands, to juniper woodland.

Moreover, I wonder whether the term "monoculture woodlands" may cause confusion, as it may be associated with plantation forestry.

L189-192: it is not entirely clear why the Wildfire category does not include fires occurring at low wind speeds. I can see the arguments for limiting the occurrence of "Prescribed Fires" to a certain range of wind speeds, but wonder on which basis the occurrence of wildfires at low wind speeds is excluded. In other words: why not include fires occurring at low wind speeds in the wildfire category and modify the text and Figures 2-4.

L192: could you clarify which smoothing function you used and show in Figure 2 the actual non smoothed values (for instance as empty circles) ?

L235-255: (Following my suggestion to L189-192) To my humble understanding, you are assuming that wildfires never occur at low wind speeds, which explains the categorisation into "prescribed fires" and "wildfires". By contrast, you could take another, eventually more balanced viewpoint to argue that fires occurring at low wind speeds (such as prescribed fires) have lower spot fire distances. Further, I might be wrong, but think that lower spot-fire distances in less-encroached landscapes may reduce the risk of fire spread, but I wonder if they would reduce the risk of fire occurrence.

Caption to Figure 1: could you clarify what land-cover types are included in the "non-receptive fuels" category?

Figure 1: an inset showing the location of Nebraska in N-America may be useful.

Figure 2: the yellow and orange lines seem to intercept the y-axis at slightly negative values, thus at negative maximum spotting distance.

Figure 4: could you highlight in Figure 1 (or in Figure 4 Column 1) the single burn unit that is shown in Column 2 ?

We look forward to receiving your revised manuscript.

Kind regards,

Walter Finsinger, PhD

Academic Editor

PLOS ONE

https://journals.plos.org/plosone/s/fileid=ba62/PLOSOne_formatting_sample_title_authors_affiliations.pdf.

“Funding for this research was provided by Nebraska Game & Parks Commission (W-125-R-1), Nebraska Environmental Trust (20-139-2), and the University of Nebraska’s Institute of Agriculture and Natural Resources. We thank landowners in the Loess Canyons who provided their prescribed burn information.”

“DT received funding

Nebraska Game & Parks Commission (W-125-R-1) http://outdoornebraska.gov/

Nebraska Environmental Trust (20-139-2) https://environmentaltrust.nebraska.gov/

University of Nebraska’s Institute of Agriculture and Natural Resources https://ianr.unl.edu/

3. We note that [Figures 1 and 4] in your submission contain [map/satellite] images which may be copyrighted. All PLOS content is published under the Creative Commons Attribution License (CC BY 4.0), which means that the manuscript, images, and Supporting Information files will be freely available online, and any third party is permitted to access, download, copy, distribute, and use these materials in any way, even commercially, with proper attribution. For these reasons, we cannot publish previously copyrighted maps or satellite images created using proprietary data, such as Google software (Google Maps, Street View, and Earth). For more information, see our copyright guidelines: http://journals.plos.org/plosone/s/licenses-and-copyright.

a. You may seek permission from the original copyright holder of Figures 1 and 4 to publish the content specifically under the CC BY 4.0 license. 

Natural Earth (public domain): http://www.naturalearthdata.com/.

Reviewers' comments:

Reviewer's Responses to Questions

**Comments to the Author**

1. Is the manuscript technically sound, and do the data support the conclusions?

Reviewer #1: Partly

2. Has the statistical analysis been performed appropriately and rigorously? 

Reviewer #1: No

3. Have the authors made all data underlying the findings in their manuscript fully available?

Reviewer #1: No

4. Is the manuscript presented in an intelligible fashion and written in standard English?

Reviewer #1: Yes

5. Review Comments to the Author

Reviewer #1: This study is aiming at guiding the stakeholders to choose the prescribed burn solution instead of waiting for wildfires to occur. The study site is in a mixed vegetation covers, offering three major landscape fuels: herb dominated, mixed herb and trees, and woodland.

I noticed three majors lacks of clear and complete details in the manuscript about the vegetation, about the landscape and about the modelling choices, that I would strongly suggest to address.

The first one is about the juniper species tested here, and not available in BehavePlus model. Even if there are similar ember flux between juniper tree and the fir considered, they don’t have the same vegetative height, moisture behavior, and flammable behavior.

As you considered partly juniper parameters directly, why not implemented directly the species into the model?

Furthermore, as it is an encroachment, the juniper trees are not mature, and should probably not attain the max height. I would suggest to use the median values instead.

We miss here important details on the fire regime (frequency, fire return intervals in the three different landscapes in wildfire conditions, is the prescribed burn annual?). We also miss information on how the fire ignition is modelled (fig. 4 landscape and single). why the ignition on the single fire panels is always at the same place? Where are the fire ignitions in landscape panels?

And most of all, in the encroachment landscape, the proportion of trees in the landscape is not detailed, as well as the torching trees portion in the trees and in the landscape.

In the discussion section, the authors stated that the prescribed fires mitigate the spot fire risk associated with wildfires, but they did not test this hypothesis. Indeed, they looked at the spot fires in these two different conditions, but did not run a wildfire after a prescribed burn in the same area. The fire risk is not reduced, but the area burned and the spot distance can be if a wildfire occurs after a prescribed burn.

In general, the differences, as stated in the results and discussion section are not tested.

Here are minor comments:

L 118-120 Seasonality and frequency of prescribed fires are lacking

L 132 what proportion of the landscape represents junipers in the encroachment landscape?

A figure of Loess Canyons Fuel complex will help to understand the fuel load and possible spread.

Fig 1: Rather than a small insert of Nebraska alone, readers can gain in having the juniper tree distribution map in the continental US

Table 3. what percentage that represents in the landscape 30 torching trees?

L 176 diameter

Fig 2 this maximum spot fire distance depends on the number of juniper trees in the landscape. Please add this information somewhere

Fig 2 and 3 are repetitive. I’ll go for fig 3 only, and change the caption (the difference is not tested here, but the range of values with the median highlighted in boxplots)

L 263 how a low intensity fire can burn a juniper tree while an intense fire is required to suppress them each 1-2 years? Is frequent here use to determine more than 1 burn a year?

L276 so this is a reverse version of what you just said.

6. PLOS authors have the option to publish the peer review history of their article (what does this mean?). If published, this will include your full peer review and any attached files.

Reviewer #1: No

---

## [Author Response · Author response to Decision Letter 0]

2 Mar 2023

Response to Review: Manuscript PONE-D-22-20383

Response to Editor

E1.1: L55: am not sure as to whether the term "extreme fire" is sufficiently clear in this context. Does "extreme" refer to the fire severity?

Response: We have changed the term to high intensity fire.

E1.2: L103-104: here you mention that vegetation communities are dominated by two "alternative states: mixed grass prairie communities and monoculture woodlands consisting of mature juniper trees". However, as of L131-132 it seems that there are at least three different levels of juniper encroachment, including vegetation types ranging from grasslands, to juniper encroached grasslands, to juniper woodland. Moreover, I wonder whether the term "monoculture woodlands" may cause confusion, as it may be associated with plantation forestry.

Response: Thank you for noting this. We have edited the text to read: “Vegetation communities can be described based on a gradient of woody encroachment from un-encroached mixed grass prairie to juniper woodlands that have resulted from unchecked woody encroachment into grassland areas.” We have removed the term “monoculture woodland” to avoid confusion.

E1.3: L189-192: it is not entirely clear why the Wildfire category does not include fires occurring at low wind speeds. I can see the arguments for limiting the occurrence of "Prescribed Fires" to a certain range of wind speeds, but wonder on which basis the occurrence of wildfires at low wind speeds is excluded. In other words: why not include fires occurring at low wind speeds in the wildfire category and modify the text and Figures 2-4.

Response: We have updated the manuscript to include all windspeeds in the wildfire category. The text has been correspondingly updated throughout, and we have edited Figures 3-4 to represent this change as recommended. Based on comment R1.10 from Reviewer 1, we have removed Figure 2 from the manuscript.

E1.4: L192: could you clarify which smoothing function you used and show in Figure 2 the actual non smoothed values (for instance as empty circles)?

Response: Based on comment R1.10 from Reviewer 1, we have removed Figure 2 from the manuscript.

E1.5: L235-255: (Following my suggestion to L189-192) To my humble understanding, you are assuming that wildfires never occur at low wind speeds, which explains the categorization into "prescribed fires" and "wildfires". By contrast, you could take another, eventually more balanced viewpoint to argue that fires occurring at low wind speeds (such as prescribed fires) have lower spot fire distances. Further, I might be wrong, but think that lower spot-fire distances in less-encroached landscapes may reduce the risk of fire spread, but I wonder if they would reduce the risk of fire occurrence.

Response: As suggested in E1.3, we have updated the manuscript to discuss wildfires as referring to the full range of wind speeds. We have updated the phrasing of this paragraph to reflect this change as suggested. 

E1.6: Caption to Figure 1: could you clarify what land-cover types are included in the "non-receptive fuels" category?

Response: We removed the receptive and non-receptive fuels map from Fig 1 while revising to accommodate E1.7 and R1.6.

E1.7: Figure 1: an inset showing the location of Nebraska in N-America may be useful.

Response: Completed.

E1.8: Figure 2: the yellow and orange lines seem to intercept the y-axis at slightly negative values, thus at negative maximum spotting distance.

Response: Based on comment R1.10 from Reviewer 1, we have removed Figure 2 from the manuscript.

E1.9: Figure 4: could you highlight in Figure 1 (or in Figure 4 Column 1) the single burn unit that is shown in Column 2?

Response: We have highlighted the burn unit in Figure 4.

 

Response to Reviewer 1

R1.1: This study is aiming at guiding the stakeholders to choose the prescribed burn solution instead of waiting for wildfires to occur. The study site is in a mixed vegetation covers, offering three major landscape fuels: herb dominated, mixed herb and trees, and woodland.

I noticed three majors lacks of clear and complete details in the manuscript about the vegetation, about the landscape and about the modelling choices, that I would strongly suggest to address.

The first one is about the juniper species tested here, and not available in BehavePlus model. Even if there are similar ember flux between juniper tree and the fir considered, they don’t have the same vegetative height, moisture behavior, and flammable behavior. As you considered partly juniper parameters directly, why not implemented directly the species into the model?

Furthermore, as it is an encroachment, the juniper trees are not mature, and should probably not attain the max height. I would suggest to use the median values instead.

Response: There is no juniper model available to use to model spot fire distance and to our knowledge, the data needed to develop our own model is not available. Thus, we chose the best option available to be input into Albini’s spot fire models.

We have clarified in the text that we selected surface fire models in an encroached grassland over the torching tree scenario to represent spot fire distances in encroached grasslands because, while similar, this model had slightly higher spot fire distances at high wind speeds and we were interested in maximum potential spot fire distances. Decreasing the tree height as suggested only decreases the spot fire distance from the torching tree, and thus, further supports our use of the surface fuel model to determine the maximum spot fire distance possible under this scenario. 

R1.2: We miss here important details on the fire regime (frequency, fire return intervals in the three different landscapes in wildfire conditions, is the prescribed burn annual?). We also miss information on how the fire ignition is modelled (fig. 4 landscape and single). why the ignition on the single fire panels is always at the same place? Where are the fire ignitions in landscape panels? And most of all, in the encroachment landscape, the proportion of trees in the landscape is not detailed, as well as the torching trees portion in the trees and in the landscape.

Response: We have clarified in the text that we do not assess fire regime, only fire behavior predicted under each listed scenario. Behave Plus is based on a point system where conditions are assumed uniform. Thus, there is no spatial configuration associated with ignition. We have added this to the text. Figure 4 (now Figure 3- see our response to R1.10) simply visualizes the different maximum spot fire distances possible based on the fuel model outputs from different wind speeds and encroachment scenarios relative to burn unit perimeters used in the Loess Canyons. We have updated the language in the Figure caption to clarify this point.

R1.3: In the discussion section, the authors stated that the prescribed fires mitigate the spot fire risk associated with wildfires, but they did not test this hypothesis. Indeed, they looked at the spot fires in these two different conditions, but did not run a wildfire after a prescribed burn in the same area. The fire risk is not reduced, but the area burned and the spot distance can be if a wildfire occurs after a prescribed burn. In general, the differences, as stated in the results and discussion section are not tested.

Response: Thank you for this comment. We agree. We re-phrased sections of the abstract, results, and discussion to reflect our findings more clearly. We have also adjusted the title of our manuscript for further clarity. We think this will also bring more clarity to the questions raised by the reviewer in R1.2.

R1.4: L 118-120 Seasonality and frequency of prescribed fires are lacking

Response: As noted in the text, burns occur between early February and late April. The landscape is currently under reclamation; thus, burns are not currently being repeated. We have added this to the text. Also, please see our response to R1.2.

R1.5: L 132 what proportion of the landscape represents junipers in the encroachment landscape? A figure of Loess Canyons Fuel complex will help to understand the fuel load and possible spread.

Response: Fuel complex percentages in the Loess Canyons are summarized in Table 1. We have added a fuel complex map for the Loess Canyons to the supporting information (S1 Fig). Changes in tree cover in the Loess Canyons can be seen on the Rangeland Analysis Platform (https://rangelands.app/). Due to copyright limitations associated with PLOS’s open access policy, we cannot provide maps from the Rangeland Analysis Platform in our manuscript. However, the reviewer can view the Loess Canyons region by going to the coordinates (40.8611304, -100.2479634). 

R1.6: Fig 1: Rather than a small insert of Nebraska alone, readers can gain in having the juniper tree distribution map in the continental US

Response: We have added a map inset of the United States to show where the Loess Canyons landscape falls and have color coded states based on the current distribution of Juniperus virginiana. 

R1.7: Table 3. what percentage that represents in the landscape 30 torching trees?

Response: We selected 30 torching trees because it was the highest number of trees we could select in Behave Plus and thus would represent the densest woodland on the landscape. Data isn’t available to calculate the number of trees per unit area in the Loess Canyons. That said, using the Rangeland Analysis Platform, we have calculated that ~9% of the Loess Canyons has greater than 50% tree cover in 2020. Tree cover in the Loess Canyons can be seen on the Rangeland Analysis Platform (https://rangelands.app/). Due to copyright limitations associated with PLOS’s open access policy, we cannot provide a map. However, the reviewer can view the Loess Canyons region by going to the co-ordinates (40.8611304, -100.2479634). 

R1.8: L 176 diameter

Response: Thank you for catching this. Fixed.

R1.9: Fig 2 this maximum spot fire distance depends on the number of juniper trees in the landscape. Please add this information somewhere

Response: Based on the reviewer’s comment in R1.10, we have removed Figure 2 from the manuscript.

R1.10: Fig 2 and 3 are repetitive. I’ll go for fig 3 only, and change the caption (the difference is not tested here, but the range of values with the median highlighted in boxplots)

Response: We have removed Fig 2 from the manuscript as recommended.

R1.11: L 263 how a low intensity fire can burn a juniper tree while an intense fire is required to suppress them each 1-2 years? Is frequent here use to determine more than 1 burn a year?

L276 so this is a reverse version of what you just said.

Response: Seedlings can be killed with low intensity fire but adult trees cannot. High intensity fire is needed to overcome adult juniper mortality thresholds. In the Loess Canyons, this is achieved through fuel manipulation. Following high intensity fire, juniper encroachment can rapidly begin again. Thus, frequent low intensity prescribed fires are needed to maintain grasslands following restoration with high intensity fire. We clarify this in the Discussion and note that fuel manipulation is used to achieve high intensity fire under prescribed fire conditions in the Loess Canyons in the Methods.

---

## [Decision Letter · Decision Letter 1]

20 Mar 2023

Spot-fire distance increases disproportionately for wildfires compared to prescribed fires as grasslands transition to Juniperus woodlands

PONE-D-22-20383R1

Dear Dr. Donovan,

We’re pleased to inform you that your manuscript has been judged scientifically suitable for publication and will be formally accepted for publication once it meets all outstanding technical requirements.

Kind regards,

Walter Finsinger, PhD

Academic Editor

PLOS ONE

Additional Editor Comments (optional):

Please make sure all data underlying the findings in their manuscript fully available (see https://journals.plos.org/plosone/s/data-availability) and that there's a link from your paper to the repository holding the data.

Reviewers' comments:

Reviewer's Responses to Questions

**Comments to the Author**

1. If the authors have adequately addressed your comments raised in a previous round of review and you feel that this manuscript is now acceptable for publication, you may indicate that here to bypass the “Comments to the Author” section, enter your conflict of interest statement in the “Confidential to Editor” section, and submit your "Accept" recommendation.

Reviewer #1: All comments have been addressed

2. Is the manuscript technically sound, and do the data support the conclusions?

Reviewer #1: Yes

3. Has the statistical analysis been performed appropriately and rigorously? 

Reviewer #1: Yes

4. Have the authors made all data underlying the findings in their manuscript fully available?

Reviewer #1: No

5. Is the manuscript presented in an intelligible fashion and written in standard English?

Reviewer #1: Yes

6. Review Comments to the Author

Reviewer #1: (No Response)

7. PLOS authors have the option to publish the peer review history of their article (what does this mean?). If published, this will include your full peer review and any attached files.

Reviewer #1: **Yes: **Berangere Leys

---

## [Editor Report · Acceptance letter]

3 Apr 2023

PONE-D-22-20383R1 

Spot-fire distance increases disproportionately for wildfires compared to prescribed fires as grasslands transition to *Juniperus* woodlands 

Dear Dr. Donovan:

I'm pleased to inform you that your manuscript has been deemed suitable for publication in PLOS ONE. Congratulations! Your manuscript is now with our production department. 

Kind regards, 

on behalf of

Dr. Walter Finsinger 

Academic Editor

PLOS ONE